# Listening to Voices from African American Communities in the Southern States about COVID-19 Vaccine Information and Communication: A Qualitative Study

**DOI:** 10.3390/vaccines10071046

**Published:** 2022-06-29

**Authors:** Ran Zhang, Shan Qiao, Brooke W. McKeever, Bankole Olatosi, Xiaoming Li

**Affiliations:** 1Department of Health Promotion Education and Behavior, Arnold School of Public Health, South Carolina SmartState Center for Healthcare Quality, University of South Carolina, Columbia, SC 29208, USA; rz1@email.sc.edu (R.Z.); xiaoming@mailbox.sc.edu (X.L.); 2School of Journalism and Mass Communications, College of Information and Communications, Prevention Research Center, Arnold School of Public Health, University of South Carolina, Columbia, SC 29208, USA; brookew@sc.edu; 3Department of Health Services Policy and Management, Arnold School of Public Health, University of South Carolina, Columbia, SC 29208, USA; olatosi@mailbox.sc.edu

**Keywords:** COVID-19 vaccines, African American, misinformation, health communication, qualitative study, USA

## Abstract

The high uptake of COVID-19 vaccines is one of the most promising measures to control the pandemic. However, some African American (AA) communities exhibit vaccination hesitancy due to mis- or disinformation. It is important to understand the challenges in accessing reliable COVID-19 vaccine information and to develop feasible health communication interventions based on voices from AA communities. We conducted 2 focus group discussions (FGDs) among 18 community stakeholders recruited from 3 counties in South Carolina on 8 October and 29 October 2021. The FGDs were conducted online via Zoom meetings. The FGD data were managed and thematically analyzed using NVivo 12. Participants worked primarily in colleges, churches, and health agencies. We found that the challenges of accessing reliable vaccine information in AA communities primarily included structural barriers, information barriers, and a lack of trust. Community stakeholders recommended recruiting trusted messengers, using social events to reach target populations, and conducting health communication campaigns through open dialogue among stakeholders. Health communication interventions directed at COVID-19 vaccine uptake should be grounded in ongoing community engagement, trust-building activities, and transparent communication about vaccine development. Tailoring health communication interventions to different groups may help reduce misinformation spread and thus promote vaccination in AA communities in the southern states.

## 1. Introduction

As of 2 March 2022, the coronavirus disease (COVID-19) pandemic has caused more than 78.9 million cases and 950,112 deaths in the United States (US) [1]. High uptake of the COVID-19 vaccine is one of the most promising measures to reduce the disease burden and control the pandemic. However, current vaccination rates in the US are suboptimal, with 67.9% of the population aged five years and older fully vaccinated [2], falling short of the objectives of the Centers for Disease Control and Prevention’s (CDC) Vaccinate with Confidence Strategy [3]. By December 2021, unvaccinated individuals had a four-times-higher risk of testing positive for COVID-19 and a fifteen-times-higher risk of dying from COVID-19 compared to fully vaccinated individuals [4]. Although this solid evidence confirms the efficacy of vaccines in preventing severe clinical outcomes of COVID-19 infection, a considerable proportion of the US population hesitates to be fully vaccinated, and this issue is relatively prevalent in minority groups, such as people of color.

People of color bear a significant burden of COVID-19 cases, hospitalizations, and deaths compared to Whites in similar age groups, but they still lag behind Whites in vaccination rates [5]. Disparities in vaccine uptake are attributed to multiple dimensions of structural inequality. Given historical and contemporary health care injustices, the African American (AA) population may not readily accept the COVID-19 vaccine as efficacious, safe, or accessible [6,7]. Moreover, recent studies have shown that misconceptions about the vaccine, mistrust in the health care system, and a lack of access to health services may discourage people from getting vaccinated, especially in communities of color [5,8,9,10,11]. The CDC and other health agencies have made great efforts to increase access to COVID-19 vaccines. However, mis- and disinformation regarding COVID-19 vaccines remains prevalent, contributing to vaccine concerns among AA communities, which in turn causes vaccine hesitancy and impedes vaccine uptake in the AA population. Research suggests that the spread of misinformation about the COVID-19 vaccine may be an important driver of vaccine hesitancy, whereas conversely, exposure to reliable medical information may increase vaccine acceptance [10,12,13]. Hence, efforts reducing the spread of misinformation and promoting access to reliable information are warranted for vaccination promotion in AA communities.

Effective health communication is essential to mitigate the negative impacts of mis- and disinformation, deliver accurate messages to the public, and promote vaccine uptake in the target audience [14]. Health communication is an application of communication concepts and theories to health-related interactions and processes that occur between individuals to improve health [15]. Health messages can influence psychological beliefs that can motivate individuals to engage in specific health behaviors [16]. Health communication can be strengthened by using efficient campaign strategies and social marketing [17] for reaching target populations and influencing voluntary behaviors to further improve health disparities for individuals and their communities [18,19]. Thus, health communication interventions can help eliminate misinformation and increase confidence in vaccination in AA communities [20]. Many efforts have been undertaken by health departments, agencies, and the CDC to develop, pilot test, and implement various health communication interventions to promote vaccine uptake [21,22,23,24].

To better tailor health communication to AA communities, it is important to understand the challenges that individuals in AA communities encounter in accessing reliable vaccine information and to develop promising strategies based on voices from these communities. Current effective health communication interventions specifically targeting AA communities remain to be understood. Therefore, the purpose of this study was to explore barriers to accessing reliable COVID-19 vaccine information in AA communities and to identify strategies recommended by community stakeholders for implementing vaccination health communication interventions.

## 2. Methods

### 2.1. Study Design and Data Collection

We conducted a qualitative study with community stakeholders by holding two focus groups discussions (FGDs) on 8 October and 29 October 2021. Each FGD had eight to ten participants to discuss access barriers to reliable COVID-19 vaccine information in AA communities in South Carolina (SC) and to identify their recommendations on health communication interventions. Eligibility criteria for recruitment included living in SC, being 18 years old and older, being a part of AA communities, and being identified as a community stakeholder by our local partner, the South Carolina Community Health Worker Association. A total of 18 community stakeholders stratified by age (i.e., ≤30 years old vs. >30 years old) were recruited from Richland, Orangeburg, and Bamberg Counties in SC (Table 1). They included leaders of churches, health agencies, community organizations, and youth ambassadors from colleges who had been actively engaged in community-based health promotion activities. Both FGDs were conducted online via Zoom meetings [25]. The facilitator explained the purpose of the study prior to each FGD. Online informed consent was signed by each participant at the beginning of the FGDs. The discussions were recorded with the consent of the participants. The study was approved by the Institutional Review Board at the University of South Carolina. Each FGD lasted approximately one hour, and the facilitator moderated the group discussions. Two researchers took field notes during each FGD. Upon completion of the FGD, each participant received a $50 gift card as compensation for their time.

According to the recommended focus group methodology [26], the University of South Carolina and the South Carolina Community Health Worker Association (SCCHWA) teams developed and modified a discussion guide (Table 2). The discussion guide aimed to identify socio-behavioral drivers of vaccine uptake and understand barriers to accessing reliable COVID-19 vaccine information in AA communities in SC. The specific objectives of the FGDs included understanding: (1) reasons for hesitancy related to COVID-19 vaccines; (2) typical misinformation and misconceptions about the vaccine; (3) facilitators and barriers to accessing reliable information; and (4) recommendations for health communication interventions to promote vaccination. During the FGDs, facilitators followed the discussion guide but asked probing questions to gain more in-depth information about the topic and to control the dynamics of the discussion for open communication among community stakeholders.

### 2.2. Data Analysis

FGDs were virtually recorded using Zoom [25] and transcribed verbatim using Otter.ai [27]. The transcripts were reviewed and edited to ensure accuracy. Qualitative data analysis software NVivo 12 [28] was used to manage and analyze FDG data. A thematic analysis approach [29] was used for data analysis, which included querying the most frequently used phrases and expanding the search to include the context of the phrases. Coding was performed by two researchers for comparison and agreement on the most significant themes. All disagreements in coding were agreed upon through discussion and reviewing the transcripts again.

## 3. Results

A total of 18 community stakeholders participated in the study, out of which 5 were males and 13 were females. These participants worked primarily in colleges, churches, and health agencies. According to the FGDs, all participants shared barriers to accessing reliable information in their communities and provided recommendations for future health communication. Barriers to accessing reliable information, including structural barriers, informational barriers, and a lack of trust, have led to high levels of COVID-19 vaccine hesitancy in AA communities. Current strategies to disseminate vaccine information and promote vaccination varied within participants’ communities and can be categorized as: recruiting trusted messengers, reaching out to target populations, and conducting health communication campaigns. The excerpts of key themes/subthemes are listed in Table 3.

### 3.1. Barriers to Accessing Reliable Information

Barriers to accessing reliable information derive from structural factors, such as the historical influence of the stigma against AA communities. Two participants emphasized the barriers created by historical reasons, mentioning the stigma and fear that the Tuskegee experiment brought to AA communities. For example, one of the participants said, “AA people have stigma about being guinea pigs due to the effects of the Tuskegee experiment, and they still need time to build their trust back”. In addition, another participant mentioned that initial financial incentives made some people less trusting of the vaccine, thus creating a barrier. Some believe there is a conspiracy to try to convince people of misinformation, and that the money given for vaccination is meant to entice people to get a microchip implanted. “Everybody is going to run with their own story, so no one is really getting a concrete message…heard one thing about the message that they’re putting a chip in your body, they’re monitoring you…”, said the participant.

Along with structural barriers, information barriers were also present, with overwhelming, unclearly explained, and inconsistent information from multiple sources, making people feel confused and exhausted. One participant said, “The main reason (for not getting vaccinated) is the whole concept [vaccine uptake] is not explained clearly to the community. There is more harm if you don’t take it, and this needs more explanation (risk/benefit assessment)”. Younger participants said, “People don’t have true education about (COVID-19 vaccines), a lot of people don’t really know the true science behind the vaccine and how it works. People are afraid of things they don’t understand”. Moreover, misinformation and conspiracy theories that are widespread on social media pose a significant challenge for young adults to get reliable information. One participant mentioned that the younger generation spends a lot more time on social media than traditional media (e.g., newspapers, radio, and TV). The younger generation is attracted to more entertaining social media, which makes them more susceptible to misinformation.

Uncertainty about the status quo due to information barriers also creates distrust of healthcare providers and government authorities. One participant said, “When they told everyone that you can go from wearing a mask, you didn’t have to wear it anymore, and then they came back and said oh well, you do need to wear it again. I think people started losing trust in the powers … the CDC are different government entities, people started to question if they really knew what they were doing. I think that kind of created barriers”. There was also a concern that politics is driving the process, making it increasingly difficult to know who to trust for reliable scientific information about vaccines. One participant said, “Even in politics as well, people are receiving so much information that it’s sometimes hard to digest and find out where the truth really lies”.

### 3.2. Suggestions for Recruiting Trusted Messengers

Some participants suggested involving churches in the promotion to provide accurate information and having pastors lead discussions to make people understand the safety of vaccines. For example, one participant said, “We need pastors in the community to have a discussion, open up their churches to tell the story that needs to be told, and help people understand the vaccine is safe”. Another participant specifically mentioned AA communities, saying “Because most AA people likely go to church and they do listen to their pastor…even host a meeting monthly, you will get a lot of participation information”. However, two participants mentioned that churches and pastors may have uncertainty in delivering messages because they may hold opposite opinions on vaccination, saying, “A lot of Christians are unvaccinated because they’re saying that the vaccine is the mark of the beast”. If similar vaccine promotion campaigns are held in churches, it is important to first confirm that the pastor’s perspective is consistent with the main goal of the campaigns. In addition, the younger group hardly goes to church now, and one young participant said, “The church approach is only fitting for those who are still physically in that building, I think my generation we’re not”.

Other participants mentioned the influence of community stakeholders who are trusted by local people to facilitate the delivery of vaccine information. People who can validate messages were recommended. One participant said, “Those (trusted people in the community) are the people that you really educate and utilize them to deliver the message… people you trusted in your community will be your best source to validate the message”. However, some participants felt that community stakeholders may not be the best persons to deliver the message, preferring to have trained professionals use data and factual information to get people to believe in vaccines, saying, “Person who is a trained professional can actually give out information…if I take this vaccine I find out that there is a 90% success rate that I won’t have any significant things happened to me, and I might be more trusting to take the vaccine”. Other potential trusted messengers include elders in the community and people who had experienced COVID-19 themselves. Younger participants mentioned that faculty, coaches, peers, and student leaders could serve as effective messengers. One young participant reported, “Students are more receptive to experts and can have a professional person talk to a mass of students”.

### 3.3. Suggestions for Reaching Out to Target Populations

Participants suggested reaching out to the target population through social events, behavioral economics, storytelling, and media strategies. Nearly half of the participants said that homecoming events, football games, and basketball games are good opportunities to reach the target group. Some participants said, “Alumni, people who come back are from the community, and they come to the homecoming. Setting up information centers where we could share vaccine information, just where we are, as a state or as a city with COVID-19”. When people are in a stadium, one participant said, “When people see things about no social distancing or masks at large public events like football games, people become discouraged about severity of disease”.

For the working population, participants said information tables or booths could be set up in organizations and companies to provide vaccination information to employees. In addition, another participant suggested reaching out to targeted populations at statewide HIV/STI conferences and pride festivals. Moreover, behavioral economics tools can help reach a broader population. Although conspiracy theories or beliefs caused difficulties in implementing financial incentives in the early stages, as information became more transparent, tailored incentives were adoptable, such as offering gift cards or grocery vouchers, providing accurate information about vaccination sites, and providing transportation. Some participants also mentioned the importance of involving churches, “Even hosting a church meeting monthly, you will get a lot of participation information”. Additionally, some participants mentioned storytelling as a powerful way to reach target groups, as “Individuals and small communities listen to people who have died in that community, so they will know, that was your neighbor that could have been you”.

Participants indicated that using various media to reach the target group would be most effective. Different age groups use different social media to get information. For example, TikTok, Instagram, and Twitter are more likely to reach younger people. People need visual information, while traditional news outlets and TV commercials can reach older people. Young participants said, “People [in] my age would be like on Instagram or TikTok or Twitter, there will be a great way to reach to people my age, but then older people of age, I would say, maybe the news and TV commercials”. Notably, one participant mentioned that we need to focus on people who do not have access to social events or computers. He suggested “Pay attention to those who don’t that socially engaged, disseminate flyers in grocery stores such as Walmart and Target”.

### 3.4. Recommendations for Health Communication

In terms of health communication, two participants suggested involving churches to provide reliable information. Another participant suggested that open dialogue with doctors could contribute to more transparent information. Regarding panel discussions and open dialogue, participants said, “Open conversation is good, especially to understand and learn more about vaccine from professional persons, but the communication are for everyday people, no jargons”. Several participants suggested involving people who have recovered from COVID-19 in health communication campaigns. They believed that storytelling is the best way to deliver the message. In health communication, it is necessary to have people who are previously infected share their experiences to emphasize the importance of accurate information and vaccine protection to the target population.

## 4. Discussion

As a new vaccine in an evolving pandemic, the COVID-19 vaccines are particularly misunderstood and, in some cases, doubted. To address this issue, effective health communication across populations is crucial to promoting vaccination [30]. This study explored barriers to accessing reliable information in AA communities and gained insights from community stakeholders on effective strategies for health communication interventions. Gaining the trust of AA communities is essential for health communication given the mistrust in the health care system due to historical factors [31]. Our results suggest that trusted messengers are important in the dissemination of accurate COVID-19 vaccine information and the promotion of vaccination behaviors in AA communities. One study showed that AAs were two to three times more likely to trust charities and religious leaders than Whites [32]. In Musa et al.’s research [33], older AAs reported significantly higher trust in informal sources of health care information (e.g., family, friends, church, and religious leaders) than Whites. Our findings are consistent with previous research findings that trusted messengers, including church members, pastors, community stakeholders, older adults, those who have experienced COVID-19, and health professionals, bear an important responsibility for disseminating information in AA communities [34,35,36]. In addition, based on the diversity of the community stakeholders involved in FGDs, we also found that student leaders, faculty, and coaches at colleges and universities can serve as key messengers to deliver vaccination messages to AA students. Information transmission and opinion formation about vaccination during COVID-19 necessitates collective learning and peer leadership, enabling individuals to solve problems through their actions [37]. In addition, individuals’ social relationships influence their vaccination opinion formation process [38,39], as individual attitudes can be changed not only by individual interactions, but also by peer and group interactions [40,41,42]. Opinion dynamics demonstrate how interactions between subpopulations holding different attitudes can result in opinion-changing processes [43]. People are more likely to interact with others who hold similar beliefs, and as a result, they are more frequently exposed to information that is consistent with their values [44]. Moreover, opinion leaders with greater influence can become central to people’s social networks, leading to information dissemination, thought change, and behavior promotion; this can happen in person and online [41,42,45]. A study of seasonal and AH1N1 influenza vaccination attitudes found that people who referred to health care providers as their source of information were more likely to be aware of the severity of influenza and to believe that vaccination was safe and effective [46]. A study about childhood vaccinations suggested pro-vaccine parents can influence those who may be vaccine hesitant [41]. Thus, guiding people through the social influence of leaders (e.g., trained personnel or peer/personal opinion leaders) can greatly facilitate crowd management during emergencies [47], similar to how pro-vaccine parents can help correct vaccine misinformation in-person or online [41,42,47]. Such messenger-led and peer-endorsed health communication interventions are simple and efficient in design, can give voice to the science for all stages from vaccine development to vaccination, and have the desired impact on the larger population.

It is worth noting that multiple sources of information, ranging from official websites to various social media platforms, may provide conflicting information, leading to confusion [48]. Similarly, in our FGDs, we found that some groups such as older adults, professionals, and college students relied on social media, news reports, and discussions among family and friends as platforms and channels for information about the COVID-19 vaccine. In addition, people that relied on less reliable sources of information had a higher likelihood of receiving incorrect information, which led to higher levels of vaccination hesitancy. In contrast, people that obtained information through physicians and professionals close to them had a better understanding of the COVID-19 vaccine. Therefore, it is important for public health officials to work with community stakeholders to employ open and transparent dialogue in the implementation of health communication activities to support accurate vaccine messaging that is culturally appropriate for AA communities [49]. Developing an appropriate Information Education Communication (IEC) approach to building positive attitudes toward vaccines by spreading awareness of vaccine availability, procedures, and benefits through mainstream and social media is critical to vaccine acceptance [50,51]. Additionally, it was interesting that participants mentioned incentives such as payment for vaccines as causing suspicion. Moving forward, campaigns need to be careful about incentivizing vaccines and be sure that explanation and clear communication is paramount.

One of our valuable findings is that setting up information tables at homecoming events, football/basketball games, regional conferences, and community parades were explicitly mentioned in the FGDs with community stakeholders as good opportunities to reach target populations and implement health communication campaigns. Unlike the potential threat of questionable information sources on social media platforms [52], it is more effective to provide easily accessible and reliable information where people live, work, learn, pray, play, and gather [53]. Moreover, in-person social events can reach and engage a more diverse group of people, especially those who do not use social media.

An important contribution of this study is collecting and analyzing the experiences of various community stakeholders to better understand information barriers and effective communication intervention strategies related to COVID-19 vaccination among AA communities. However, the current study has limitations. Our strategy of utilizing convenience sampling resulted in more than half of the participants being female, or between the ages of 18 and 30, or from colleges (including students and staff), which may result in bias in the results. However, these young adults who participated in the study had been engaged in community-based health promotion interventions through organizing and/or coordinating various online or offline events and activities (e.g., health communication campaign, vaccination advocacy, community health day) in their communities as junior ambassadors. They were more experienced and competent than general college students to represent their respective communities as a voice and provided constructive suggestions. Future studies could utilize stratified sampling to improve the accuracy and representativeness of the results by reducing sampling bias. Future research also needs to expand to other important stakeholders that play an important role in health communication (e.g., AA associations). In addition, further studies could be conducted among other groups to understand health communication and social media use during the pandemic.

## 5. Conclusions

The scope and challenges of COVID-19 vaccine dissemination and promotion are unprecedented, especially in AA communities. Vaccination hesitancy in AA communities is largely driven by misinformation and mistrust. Therefore, vaccine promotion interventions should be based on sustained community engagement, trust-building activities, and transparent communication about vaccine development. Health communication interventions play a particularly important role in vaccination promotion. Accurate messaging, clear communication from opinion leaders and also among peers, and behavioral interventions require community support and engagement. To address the challenges of vaccination in AA communities in the southern states, our study explored the impact of the threat of COVID-19 mis- and disinformation and barriers to accessing accurate information in AA communities, as well as how health communication interventions can be more effective from the perspective of community stakeholders. When conducting health communication interventions, we suggest strategies that use a combination of the credibility of key messengers, multi-sourcing of social media, and accessibility of social events to increase trust and confidence in vaccination in AA communities. Furthermore, tailoring health communication interventions for different groups (e.g., by age) and offering personalized and peer-led communication strategies may help reduce vaccination hesitancy, thus promoting vaccination rates in AA communities in the southern states.

## Figures and Tables

**Table 1 vaccines-10-01046-t001:** Demographic characteristics of the participants in the FGDs.

Variable	*n* (Total = 18)	%
Gender		
Male	5	28
Female	13	72
Age (years)		
18–30	12	67
31–49	4	22
50+	2	11
County		
Richland County	11	61
Orangeburg/Bamberg County	7	39
Affiliation		
College	10	55.5
Health agency	1	5.5
Church	7	39

**Table 2 vaccines-10-01046-t002:** Focus group discussion guide.

Part One: General Attitude
What are people saying in your community about COVID-19?
2. What are ways that you or people in your circle believe are the best way to protect themselves from the COVID-19 virus?
Part Two: COVID-19 Information
3. What information have you received about the vaccine?
4. Where are you getting your information about COVID-19?
5. Are there barriers to getting honest information about COVID-19?
6. What can be done to improve confidence in information regarding COVID-19?
Part Three: Promotion Strategy
7. How can we promote more transparent information about COVID-19 in your community?
8. Are there any upcoming events in your area where people can be provided information regarding COVID-19 and the vaccine?
Part Four: Media Strategy
9. What would be the most effective way to reach people–social media, TV, radio, printed materials (posters, newspapers) or word of mouth?
10. Who in your community will people more likely listen to?

**Table 3 vaccines-10-01046-t003:** Key themes and excerpts.

Main Themes	Sub-Themes		Excerpts
Barriers to accessing reliable information	Structural barriers	Historical influence of stigma against AA	“AA people have stigma about being guinea pigs due to the effects of the Tuskegee experiment, and they still need time to build their trust back.”
		Not all people get vaccinated in the first place	“Have they all been truthful in on the same page when this first started, you were to go more participation and more people interested, but they made it so difficult when this COVID first started, they were only vaccine certain people in certain areas, so they made everything so hard and difficult I didn’t get mine until like almost six months later when it wasn’t that difficult.”
	Health concerns	Having allergies	“A lot have not received the vaccine because they have allergies which cause them to fear taking the vaccine.”
		Underlying health problems	“A lot of people have not addressed that issue to people who do have other underlying problems.”
	Information barriers	Misinformation/misconception	“People are misinformed, not doing enough of their own research, and basing their knowledge on social media.” “The main reason is the whole concept isn’t being explained clearly to the community. The more harm if you don’t take it needs to be explained more (risk/benefit appraisal).”
		No consistent message	“We’re not getting one consistent message (politician, CDC, healthcare providers, the internet), which is causing mixed messages and conspiracy theories. Nowadays a lot of people get their info off the Internet, so people are running with their own story, no concrete messages. We need one voice to get one consistent message out.”
		Information transparency does not reach enough people	“They’re going to watch the wrong stuff and then they’re going to just keep circulating these false news, so I believe that the transparency out there is just that it’s not reaching enough persons, for it to be sprint.”
		Conspiracy	“Conspiracy theory is what I believe is one of the main misconception of it, like, for example in my community I’ve heard persons like oh I’m not getting the vaccine, because it causes cancer.”
		Miseducation	“Based on the information they got from social media, taking the vaccine may have worse effects than not taking the vaccines, I think it’s just a lot of miseducation and not doing your own research.”
		Loss of trust in authority	“When they told everyone that you can go from wearing a mask you didn’t have to wear it anymore, and then they came back and said Oh well, you do need to wear it again I think people started losing trust in the powers that be, as far as the CDC are different government entities people started to question if they really knew what they were doing. I think that kind of created barriers.”
		Lack of trust on social media	“Trust goes hand in hand, I think you know the social media trust has kind of been tarnished.”
Suggestions for recruiting trusted messengers	Pastors/religious leaders		“You’re gonna have to use influences. You’re going to have to utilize people they trust in the community. The church they trust that.” “Need pastors in the community to have a discussion, open up their churches to tell the story that needs to be told, and help people understand the vaccine is safe.” “Get religious leaders and other leaders to relay the messages.”
	Community leaders		“Even going to leaders within that community you’re going to have to sort out they have social influences.” “Community leaders and gatekeepers that people who hold the key to the different organizations in the areas they’re the ones that are you know are probably the most effective and getting you know results.”
	Health professionals		“Have a professional person talk to a mass of students. Students are more receptive to experts.” “Person who’s a trained professional can actually give out information to so people can be like whoa if I take this vaccine, I find out that there’s a 90% success rate that I won’t have any significant things happened to me. And I might be more trusting to take the vaccine.”
	Gatekeepers of communities/colleges		“This would be in all communities’ geographical different geographical areas where you have people there are people who are gatekeepers are of communities. We have to have those gatekeepers of those communities to have factual information and being able to provide that information to other people other communities. Even on the college level, there are gatekeepers in in the colleges, where other students that’s they believe, and they will take information from other students and maybe on faculty and those I think those are important. People in different communities that other folks listen to.”
Suggestions for reaching out to target populations	Community health day		“I consider the entire state of South Carolina my community, because I feel like I at some point during my week I’m in a portion of the state in some capacity, so I know that tomorrow my story my chapter is actually hosting a community day, where they will be giving out information about the COVID-19 vaccine. So I know that that’s something that they’re doing and where our chapter house is located. it’s in a prime location for African Americans to get that information because it’s in the midst in the middle of a predominantly African American neighborhood So hopefully we have a lot of people that come out and hear what they need to or in regards to that.”
	Homecoming event		“Homecoming will be a great time to have an event to talk about COVID-19.” “It’s homecoming season so having tent set up at a homecoming tailgate to talk to people about the vaccine that makes it real for us because that puts a face to you know, to the cause, if that makes sense.”
	Football/basketball games		“When people see things about no social distancing or masks at large public events like football games, people become discouraged about severity of disease.” “Football game and basketball game. I truly believe the information can be given out there. There are variety of people who will be vaccinated as well as a variety of people who will not be vaccinated. And that information could be given with people at the congregation of the game.”
	Statewide HIV/STI conference and the Pride festival		“My organization is offering vaccines at an HIV/STI conference and the Pride festival.”
Recommendations for health communication	Involve churches to provide reliable information		“Most Black people likely will do go to church and they do listen to her pastor so she was correct on that. They would have like you said in advance, or even host a meeting monthly, you will get a lot of participation information.”
	Open dialogue with doctors could contribute to more transparent information		“Open conversation is good, especially to understand and learn more about vaccine from professional persons, but the communication are for everyday people, no jargons.” “There needs to be a place that has a dialogue about it because without that it feels coercive.” “We can promote more transparent information by having conversations, because I think … actual doctors who have done that that that research to understand how vaccines work and understand how this whole the COVID-19 vaccine is not like the flu vaccine and just being able to have that conversation in an environment that’s not hostile I think that’s what’s missing is open conversations between two different from the two different points of views.”
	Storytelling to deliver message		“The storytelling of those who have gone to covert who has been hospitalized was suffered to tell their stories that would be the main focus on the news on these public at conferences showed the effects of covert itself, I find it better and being in the field of HIV telling your story.”

## Data Availability

The data presented in this study are available on request from the corresponding author (S.Q.). The data are not publicly available due to privacy concerns.

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
