# Peer review of "Listening to Voices from African American Communities in the Southern States about COVID-19 Vaccine Information and Communication: A Qualitative Study"

_vaccines, 2022, doi:10.3390/vaccines10071046_

Round 1

Reviewer 1 Report

I enjoyed reading the paper. I believe that it is fundamental to discuss directly with people many relevant issues related to our behavior as members of society. Vaccination against COVID-19 made several communication problems emerge clearly in all countries. It is crucial to incentivize the development of strategies allowing people's awareness of decisions and behaviors to increase.
I would encourage the authors to extend these kinds of research to other groups of people since it could advantage the understanding of the dynamics driving the formation of echo chambers in social media, as an example.
I have appreciated the conclusion regarding the diffusion of correct knowledge on vaccines (explanation and clear communication is paramount), which would advantage all of us. Moreover, I agree with the authors that the study requires further extensions to include larger population samples.

Author Response

Reviewer 1:

  1. I enjoyed reading the paper. I believe that it is fundamental to discuss directly with people many relevant issues related to our behavior as members of society. Vaccination against COVID-19 made several communication problems emerge clearly in all countries. It is crucial to incentivize the development of strategies allowing people's awareness of decisions and behaviors to increase.

Response: Thank you for positive feedback on our manuscript.
2. I would encourage the authors to extend these kinds of research to other groups of people since it could advantage the understanding of the dynamics driving the formation of echo chambers in social media, as an example.

Response: We agree with the reviewer and have added suggestions for future research that would extend this type of study to other groups of people (discussion section). In future study, we would be very interested in exploring the barriers to access accurate vaccine information and the choices of communication methods in other populations.

  1. I have appreciated the conclusion regarding the diffusion of correct knowledge on vaccines (explanation and clear communication is paramount), which would advantage all of us. Moreover, I agree with the authors that the study requires further extensions to include larger population samples.

Response: We agree, and we have addressed this issue in the discussion section as one of the study limitations. In the future study, we will conduct more focus group discussions to understand the phenomena or issues observed by community stakeholders during vaccine promotion.

Reviewer 2 Report

This is an important, interesting and well written report, suggesting how to influence vaccine acceptance among African Americans in South Carolina.   The points are all well addressed, the ideas are strong.   Many of the same ideas for influencing public health behaviors have been noted for other conditions (obesity, flu vaccine, diet) but public health seems not to have figured out how to implement some necessary changes. 

Many of your participants are from colleges, they may have learned about prior ideas for influencing a population.  You correctly acknowledge this as a limitation.

For this report, two questions. 

1) can you describe what you mean by "community leaders'.  Since many of your focus group participants are young, in what way were they 'leaders'.  

2) would it be possible to condense/ synthesize some of the text into tables, for easier reading/ more access to your main take home messages. 

Author Response

Reviewer 2:

  1. This is an important, interesting and well written report, suggesting how to influence vaccine acceptance among African Americans in South Carolina. The points are all well addressed, the ideas are strong. Many of the same ideas for influencing public health behaviors have been noted for other conditions (obesity, flu vaccine, diet) but public health seems not to have figured out how to implement some necessary changes. 

Many of your participants are from colleges, they may have learned about prior ideas for influencing a population.  You correctly acknowledge this as a limitation.

Response: Thank you for the positive feedback on the manuscript.

  1. For this report, two questions. 

1) Can you describe what you mean by "community leaders”. Since many of your focus group participants are young, in what way were they “leaders”.  

Response: Sorry for the confusion. Key community stakeholder would be a more appropriate term. The young participants have engaged in community-based health promotion interventions. They are actively involved in various types of community activities as junior ambassadors, including COVID-19 vaccine advocacy, and they are more experienced and qualified than the general college students to represent the community to express their thoughts.

2) Would it be possible to condense/synthesize some of the text into tables, for easier reading/ more access to your main take home messages? 

Response: Thank you for your suggestion. We agree, and we have added a table to make the information easier to read.

Reviewer 3 Report

This paper presents a statistical investigation on the dynamics of diffusion of information and opinion information to move a population towards the awareness od the need to start a vaccination program.

I have red with interest this paper as it tackles an important topic. However, I think that is shows some lights, but also shades. A positive feature of the article is that it tackles an important topic, i.e. opinion formation in the case of asymmetric transfer of messages. Another interesting topic is that the investigation accounts for heterogeneity of the population.

On the other hand, the shades are the following:

  • The number of individuals used for the statistics is too small to work out reliable conclusions. In addition, the filter from data to conclusions is too vague and it should be made more precise.

  • The dynamics of opinion formation is complex and several variables should be considered to extract an appropriate interpretation strategy.

It seems to me that the author should not skip over the above two topics and some detailed  reasoning should be inserted in the last part of the paper with critical analysis and reference to the pertinent bibliography

Author Response

Reviewer 3:

  1. This paper presents a statistical investigation on the dynamics of diffusion of information and opinion information to move a population towards the awareness of the need to start a vaccination program.

I have read with interest this paper as it tackles an important topic. However, I think that is shows some lights, but also shades. A positive feature of the article is that it tackles an important topic, i.e. opinion formation in the case of asymmetric transfer of messages. Another interesting topic is that the investigation accounts for heterogeneity of the population.

On the other hand, the shades are the following:

  • The number of individuals used for the statistics is too small to work out reliable conclusions. In addition, the filter from data to conclusions is too vague and it should be made more precise.
  • The dynamics of opinion formation is complex and several variables should be considered to extract an appropriate interpretation strategy.

It seems to me that the author should not skip over the above two topics and some detailed reasoning should be inserted in the last part of the paper with critical analysis and reference to the pertinent bibliography.

Response:

We appreciate the reviewer’s positive comment. However, our paper focused on a qualitative study of COVID-19 vaccine information and communication, and we did not include any variables or statistical investigation on the dynamics of diffusion of information and opinion information. We are not sure whether this comment was indeed for our paper and would appreciate further explanation.

Reviewer 4 Report

Estimated authors of the paper ""Where the truth really lies”: Listening to voices from African American communities in the Southern States about COVID-19 vaccine information and communication"

I've read and appreciated your qualitative research work on the barriers and facilitators towards SARS-CoV-2 vaccine in African American. Within the implicit limits of qualitative research, the present paper appears well reported, and its content may be particularly interesting for medical professionals working in vaccination settings.

However, some improvements are in need before the eventual acceptance.

First, I would recommend the Authors to implement a summary of the items that were initially discussed within the focus groups (necessary), and a detailed reporting of most significant statements would be appreciated, if possibile (i.e. an optional improvement).

Second, Authors should discuss in more extensive details the limits of this study, because of the sampling of authors and topics.

I'm confident that the Authors will be able to quickly improve the present paper accordingly, and I would like to read the updated paper.

Author Response

Reviewer 4:

  1. Estimated authors of the paper ""Where the truth really lies”: Listening to voices from African American communities in the Southern States about COVID-19 vaccine information and communication"

I've read and appreciated your qualitative research work on the barriers and facilitators towards SARS-CoV-2 vaccine in African American. Within the implicit limits of qualitative research, the present paper appears well reported, and its content may be particularly interesting for medical professionals working in vaccination settings.

Response: Thank you for positive feedback on our manuscript.

  1. However, some improvements are in need before the eventual acceptance.

First, I would recommend the Authors to implement a summary of the items that were initially discussed within the focus groups (necessary), and a detailed reporting of most significant statements would be appreciated, if possible (i.e., an optional improvement).

Response: We agree with the reviewer and have added a table (Table 2. Focus group discussion guide) to show the whole picture of the focus group discussions. In response to the comments from Reviewer 2, we have added a table (Table 3. Key themes and excerpts) to categorize the themes and example excerpts summarized from the statements of the participants. All detailed and most important statements are presented in the results section.

  1. Second, Authors should discuss in more extensive details the limits of this study, because of the sampling of authors and topics.

Response: We have provided a more detailed description of the limitations (in response to age, gender, and affiliation imbalance in the sample) and will address these limitations in future studies. However, we did find that young people play a significant role in promoting vaccination interventions in the community.

  1. I’m confident that the Authors will be able to quickly improve the present paper accordingly, and I would like to read the updated paper.

Response: We greatly appreciate your support. We have modified the paper based on the comments of the reviewers and would like to communicate with you further.

Reviewer 5 Report

The paper concerns COVID-19 vaccination hesitancy in some African American (AA) communities and developing effective health communication inteventions. Obviously, this problem is very important and ways of convincing various groups of people to vaccination are crucial in dealing with this and possible future pandemics.

The Authors conducted disscussions with AA community leaders from Southern California and on this basis they formulated some recommendations concerning enhancing of health communication effectiveness. These recommendations may be useful in practice but they seem to be what could be expected. In other words, in my opinion, such recommendations are quite natural and they could be formulated even without the analyzes performed by the Authors.

Moreover, there are almost no quantitative aspects of the conducted research.

In addition, as the Authors wrote, there is a serious limitation of the study, i.e., 10 of 18 community leaders are from colleges, what could bias the obtained results.

Minor point: what does it mean "the University of xxxx"?

Summarizing, in my opinion the paper should not be published in Vaccines in its present form.

Author Response

Reviewer 5:

  1. The paper concerns COVID-19 vaccination hesitancy in some African American (AA) communities and developing effective health communication interventions. Obviously, this problem is very important and ways of convincing various groups of people to vaccination are crucial in dealing with this and possible future pandemics.

Response: We appreciate the reviewer’s positive comment.

  1. The authors conducted discussions with AA community leaders from Southern Carolina and on this basis, they formulated some recommendations concerning enhancing of health communication effectiveness. These recommendations may be useful in practice, but they seem to be what could be expected. In other words, in my opinion, such recommendations are quite natural, and they could be formulated even without the analyzes performed by the Authors.

Response: We appreciate the reviewer’s positive comment. We agree that our findings did not come as a surprise or an unexpected perspective, but we believe it is necessary to understand the real barriers of AA communities. Therefore, our study provides evidence-based recommendations. These recommendations were derived by key community stakeholders based on interactions with community members, and although they can be expected, they are effective for vaccination promotion.

  1. Moreover, there are almost no quantitative aspects of the conducted research.

Response: We appreciate the reviewer’s comments. We agree that we did not include quantitative aspects because this is a qualitative study. We agree that our study analysis would have been richer if we had collected some quantitative data. However, such a study design would become a mixed method, which is beyond the scope of the current study.

  1. In addition, as the Authors wrote, there is a serious limitation of the study, i.e., 10 of 18 community leaders are from colleges, what could bias the obtained results.

Response: We agree that we have an imbalanced sample. In the College category we include both college students and staff. Both students and college staff provided different information based on their own communities. In response to the comments from Reviewer 1 and Reviewer 2, we have clarified younger participants/community stakeholders play an important role in the community. They are actively engaged in community activities (e.g., health communication campaign, COVID-19 vaccine advocacy, community health day, etc.). We would pay more attention to balance the distribution of the sample in further studies to avoid causing biases.

  1. Minor point: what does it mean "the University of xxxx"?

Summarizing, in my opinion the paper should not be published in Vaccines in its present form.

Response: This study was approved by the Institutional Review Board at the University of South Carolina.

Round 2

Reviewer 2 Report

Your revisions are appreciated.

Author Response

Thank you for positive feedback on our revisions. One of coauthors, a native speaker of English has further reviewed and checked the language and style (including grammar and spell).

Reviewer 3 Report

The authors have provided to some technical revisions which have improved the quality of the presentation. However, they have not considered the indications of the referee report which suggested to insert some statement on the fact that the overall dynamics is influeced by opinion dynamics as it happens in our society as well expressed by the visionary ideas od Herbert A. Simon, Nobel laureate in 1978. The book "The Science of Artificial" MIT Press, Boston, USA Third Edition 2019.

The answer by the author is basically "we have not considered" statistics of opinion formation and then they ask for some indications. As a reviewer, I claim that the  author should simply mention that this topic is not treated, but it should be treated. If might be as a research perspective. The related research activity in the field is reported in

https://www.worldscientific.com/doi/pdf/10.1142/S0218202521500408

NOT to be cited, but it is a source of bibliography on the field

Author Response

  1. The authors have provided to some technical revisions which have improved the quality of the presentation.

Response: Thank you for positive feedback on our revisions.

  1. However, they have not considered the indications of the referee report which suggested to insert some statement on the fact that the overall dynamics is influenced by opinion dynamics as it happens in our society as well expressed by the visionary ideas of Herbert A. Simon, Nobel laureate in 1978. The book "The Science of Artificial" MIT Press, Boston, USA Third Edition 2019.

The answer by the author is basically "we have not considered" statistics of opinion formation and then they ask for some indications. As a reviewer, I claim that the author should simply mention that this topic is not treated, but it should be treated. If might be as a research perspective. The related research activity in the field is reported in https://www.worldscientific.com/doi/pdf/10.1142/S0218202521500408 NOT to be cited, but it is a source of bibliography on the field.

Response: We really appreciate the reviewer’s comments. We have read the book and the study recommended by the reviewer. We have added opinion dynamics and opinion formation to the discussion (lines 279-297). Exposure to COVID-19 misinformation and attitudes/intentions toward vaccination in the African American community are influenced by opinion dynamics. In addition, in the context of opinion dynamics, we highlight the importance of peer leadership and opinion leaders in the process of opinion formation. The research suggested by the reviewer and other relevant studies are also cited in this manuscript.

Reviewer 5 Report

All my previous main critical comments remain valid in the context of the corrected version of the paper. Hence, it would be very difficult for me to change my general opinion abour the paper, i.e., despite that I appreciate the efforts of the Authors, in my opinion the paper in its current form is not sufficiently well-suited for publication in Vaccines.

Author Response

We appreciate the reviewer for sharing the comments and opinions. As suggested by reviewer 3, we have rewritten the discussion with opinion dynamics and opinion formation to make the information dissemination and intervention recommendations more relevant to social dynamics. We also added more context in this section to show how the COVID-19 context is similar to other vaccine contexts, including childhood vaccinations. This study is based on an ongoing vaccine promotion intervention project in African American communities, and the recommendations and analyses in the study provide a foundation for peer advocates and health professionals to better deliver interventions. We also aimed to address knowledge gaps about the lack of understanding of multiple levels of vaccination barriers and develop tailored interventions. We appreciate your thoughts, and hope the additional context provided in the revised paper will help bolster the value and relevance of the current study.

Round 3

Reviewer 5 Report

The current version of the paper is better elaborated than the initial one. Hence, in my opinion the paper could be considered for a possible publication.